# Upregulation of human GD3 synthase (*hST8Sia I*) gene expression during serum starvation-induced osteoblastic differentiation of MG-63 cells

**So-Young An[1], Hyun-Kyoung Yoon[1], Kyoung-Sook Kim[1], Hee-Do Kim[2], Jong-Hyun Cho[1], Hyeon-Jun Kim[3], Cheorl-Ho Kim[2]\*, Young-Choon Lee[1]\***

**1** Department of Medicinal Biotechnology, College of Health Sciences, Dong-A University, Busan, South Korea, **2** Molecular and Cellular Glycobiology Unit, Department of Biological Sciences, SungKyunKwan University, Kyunggi-Do, South Korea, **3** Department of Orthopaedic Surgery, College of Medicine, Dong-A University, Busan, South Korea

\* yclee@dau.ac.kr (YCL); chkimbio@skku.edu (CHK)

**Data Availability Statement:** All relevant data are within the paper and its Supporting Information files.

## Abstract

In this study, we have firstly elucidated that serum starvation augmented the levels of human GD3 synthase (*hST8Sia I*) gene and ganglioside GD3 expression as well as bone morphogenic protein-2 and osteocalcin expression during MG-63 cell differentiation using RT-PCR, qPCR, Western blot and immunofluorescence microscopy. To evaluate upregulation of *hST8Sia I* gene during MG-63 cell differentiation by serum starvation, promoter area of the *hST8Sia I* gene was functionally analyzed. Promoter analysis using luciferase reporter assay system harboring various constructs of the *hST8Sia I* gene proved that the cis-acting region at -1146/-646, which includes binding sites of the known transcription factors AP-1, CREB, c-Ets-1 and NF-κB, displays the highest level of promoter activity in response to serum starvation in MG-63 cells. The -731/-722 region, which contains the NF-κB binding site, was proved to be essential for expression of the *hST8Sia I* gene by serum starvation in MG-63 cells by site-directed mutagenesis, NF-κB inhibition, and chromatin immunoprecipitation (ChIP) assay. Knockdown of *hST8Sia I* using shRNA suggested that expressions of *hST8Sia I* and GD3 have no apparent effect on differentiation of MG-63 cells. Moreover, the transcriptional activation of *hST8Sia I* gene by serum starvation was strongly hindered by SB203580, a p38MAPK inhibitor in MG-63 cells. From these results, it has been suggested that transcription activity of *hST8Sia I* gene by serum starvation in human osteosarcoma MG-63 cells is regulated by p38MAPK/NF-κB signaling pathway.

## Introduction

It has been reported that serum starvation induces cell differentiation in various cells such as human prostate cancer LNCaP cell [1], mouse embryonal carcinoma cell [2], rat proximal tubule cell [3], mouse neuroblastoma Neuro2a cell [4], and C2C12 myoblast cell [5]. The

**Funding:** This work was supported by the Dong-A University research fund. Article processing charge will be paid by the DAU fund. The funder of Dong-A University research fund had no role in study design, data collection and analysis, decision to publish, or preparation of the manuscript.

**Competing interests:** The authors declare no conflict of interest.

human osteosarcoma MG-63 cell line has been used as representative subpopulation of osteoblasts which are osteoblast precursors or early undifferentiated osteoblast-like cells. MG-63 cells synthesize bone morphogenic protein-2 (BMP-2) and osteocalcin as markers of osteoblastic differentiation [6–10]. The gene expressions of BMP-2 and osteocalcin were increased by serum starvation in MG-63 cells [9, 10].

Among human sialyltransferases, seven enzymes including hB4Gal T I, hB3GalT IV, hST3Gal II, hST3Gal III, hST3Gal V, hST8Sia I and hST8Sia V are involved in biosynthetic pathway of the gangliosides [11, 12]. Our previous studies have shown that serum starvation also triggers cell differentiation of human MG-63 osteoblastic cells [9, 10]. Moreover, we found that during MG-63 cell differentiation, serum starvation increased gene expressions of the human β-galactoside α2,6-sialyltransferase (*hST6Gal I*) [10], as well as the human GM3 synthase (*hST3Gal V*), also called lactosylceramide α2,3-sialyltransferase, which catalyzes biosynthesis of ganglioside GM3 [9]. Interestingly, during our studies on serum starvation-induced expression of human sialyltransferases, expression of *hST8Sia I* gene was particularly increased, being the highest level. Based on these findings, we continued to investigate the gene expression of ganglioside-specific sialyltransferases during osteoblast differentiation.

In the present study, the expression of hST8Sia I, known as Sia:GM3 α2,8-sialyltransferase, which catalyzes the formation of disialoganglioside GD3 from monosialoganglioside GM3, as a key biosynthetic enzyme of the b- and c-series gangliosides [11, 12], specifically increases during the differentiation of MG-63 cells by serum starvation. Furthermore, the distinct elevation of hST8Sia I protein level was observed in serum starvation-induced MG-63 cells. In parallel with the *hST8Sia I* gene, we examined whether serum starvation alters in the expression of differentiation marker molecules of BMP-2 and osteocalcin at the mRNA levels in MG-63 cells. In conclusion, we have here elucidated transcriptional upregulation of *hST8Sia I* gene responsible for the increment of its gene expression induced by serum starvation.

## Materials and methods

### Cell cultures

Cell culture of human MG-63 and U2OS osteoblastic cells was conducted as described previously [9, 10] using Dulbecco's modified Eagle's medium (DMEM; WelGENE Co., Daegu, Korea) supplemented with 100 U/ml of penicillin, 100 μg/ml of streptomycin, and 10% (v/v) fetal bovine serum (FBS) (Gibco BRL, Life Technologies; Grand Island, NY, USA). To induce cell differentiation and the increased expression of hST8Sia I gene, the cell was placed in FBS-free medium for different time periods.

### Reverse transcription-polymerase chain reaction (RT-PCR) and quantitative real-time PCR (qPCR)

Isolation of total RNA from cultured cells, first-strand cDNA synthesis, and PCR amplification were performed as previously described [9, 10]. Six kinds of ganglioside-specific human sialyltransferase genes were amplified by PCR condition with specific primers shown in S1 Table. Time-dependent RT-PCR of specific hST3Gal V, bone morphogenic protein-2 (BMP-2), osteocalcin and β-actin was conducted with primers shown in Table 1. PCR products were analyzed by 1% agarose gel electrophoresis. Real-time qRT-PCR was performed as previously described [9, 10] using a CFX96™ Real-Time system with SYBR Premix (Bio-Rad) and the specific primers as shown in Table 1. Target gene expression was normalized to β-actin mRNA expression.

**Table 1. Primer sequences used for RT-PCR, qPCR and ChIP in this study.**

| Primer | Sequence | Strand | Purpose |
|---|---|---|---|
| hST8Sia I | 5'– TGTGGTCCAGAAAGACATTTGTGGACA –3' | Sense | RT-PCR |
| hST8Sia I | 5'– TGGAGTGAGGTATCTTCACATGGGTCC –3' | Antisense | |
| BMP-2 | 5'– ATGTTCGCCTGAAACAGAGACCCA –3' | Sense | RT-PCR |
| BMP-2 | 5'– CTTACAGCTGGACTTAAGGCGTTTC –3' | Antisense | |
| Osteocalcin | 5'– ATGAGAGCCCTCACACTCCTC –3' | Sense | RT-PCR |
| Osteocalcin | 5'– GCCGTAGAAGCGCCGATAGGC –3' | Antisense | |
| β-actin | 5'– CAAGAGATGGCCACGGCTGCT –3' | Sense | RT-PCR |
| β-actin | 5'– TCCTTCTGCATCCTGTCGGCA –3' | Antisense | |
| hST8Sia I | 5'– TTCAACTTACTCTCTCTTCCCACA –3' | Sense | qPCR |
| hST8Sia I | 5'– TCTTCTTCAGAATCCCACCATT –3' | Antisense | |
| BMP-2 | 5'– GGGTTGGAACTCCAGACTGT –3' | Sense | qPCR |
| BMP-2 | 5'– GAAGAGTGAGTGGACCCCAG –3' | Antisense | |
| Osteocalcin | 5'– GAGGGCAGCGAGGTAGTGAA –3' | Sense | qPCR |
| Osteocalcin | 5'– GGCTCCCAGCCATTGATACA –3' | Antisense | |
| ALP | 5'– CCACGTCTTCACATTTGGTG –3' | Sense | qPCR |
| ALP | 5'– AGACTGCGCCTAGTAGTTGT –3' | Antisense | |
| β-actin | 5'– ACCCACTCCTCCACCTTTGAC –3' | Sense | qPCR |
| β-actin | 5'– CCTGTTGCTGTAGCCAAATTCG –3' | Antisense | |
| hST8Sia I | 5'– CTCCGCCACACTCAGGGACT –3' | Sense | ChIP |
| hST8Sia I | 5'– ACAAACGCCCGGGGATTG –3' | Antisense | |

## Transfection and luciferase assay

The luciferase reporter plasmids (pGL3-1146/-646 to pGL3-2646/-646) and mutant plasmids with base substitutions in the CREB, AP-1, c-Ets-1, NF-κB binding sites, have been described elsewhere [13–18]. Transfection and luciferase assay to evaluate hST8Sia I promoter activity by serum starvation were carried out as previously described [10]. MG-63 cells were plated at a density of $5.0 \times 10^4$ cells/well in 24-well culture plate. After incubating for 15 h, cells were co-transfected with 0.5 μg of luciferase reporter plasmid and 50 ng of pRL-TK as the control *Renilla* luciferase vector (Promega; Madison, WI, USA), using 1 μl Lipofectamine 2000 (Invitrogen). After 18 h incubation in normal medium with serum, the medium was replaced with serum-free medium. Cells incubated for an additional 24 h in serum-free medium were harvested and assayed using the Dual-Luciferase Reporter Assay System (Promega) and a Glo-Max$^{Tm}$ 20/20 luminometer (Promega).

## Immunofluorescence staining

Immunofluorescence staining was conducted as previously described [18]. MG-63 cells grown for 24 h on sterile coverslips containing medium with or without 10% FBS were fixed with 4% paraformaldehyde for 10 min at 37°C, washed three times with PBS, and blocked with 5% BSA for 1 h at 37°C. After incubation for overnight at 4°C with the anti-GD3 monoclonal antibody (mAb) (mouse IgM, Kappa-chain, clone, GMR19; Seigakagu, Tokyo, Japan), cells were reacted with fluorescein isothiocyanate (FITC)-conjugated anti-mouse IgG/M/A mixture (Vector labs, F1-1000) used as the secondary antibody for 1 h at 37°C. The nucleus was stained with DAPI for 10 min at room temperature. Fluorescence images of cells were obtained by using LSM 700 confocal laser scanning microscope (Carl Zeiss, Oberkochen, Germany).

## Western blot analysis

Western blot analysis was conducted as described previously [9]. The cell lysates obtained by lysis in RIPA buffer were separated on SDS-polyacrylamide gels and transferred to PVDF membrane. Membrane was incubated with sequentially with hST8Sia I-specific antibody (#sc-46982, Santa Cruz, CA) and horseradish peroxidase (HRP)-conjugated secondary antibody (Enzo Life Science, Farmingdale, NY). GAPDH (#sc-20357, Santa Cruz, CA) was used as an internal control. Blots were detected using the ECL chemiluminescence system (GE Healthcare, Piscataway, NJ, USA).

## Chromatin immunoprecipitation (ChIP) assay

ChIP assay was performed using a ChIP assay kit (Millipore, USA) according to the manufacturer's instruction. As described previously [16–18], cells were cross linked in 1% formaldehyde at 37°C for 10 min to cross-link DNAs and protein, and then sonicated to shear genomic DNAs to average size of 200–1000 bp. Immunoprecipitation was carried out using 4 μg of NF-κB antibody (Santa Cruz Biotechnology) and IgG antibody (Sigma) as negative control. The purified ChIP DNA or input DNA was used for PCR analysis using primers flanking NF-κB binding site (-731/-722) on the hST8Sia I promoter (Table 1).

## Knockdown of hST8Sia I

Lentiviral plasmid pLKO.1 containing shRNA targeting hST8Sia I (TRCN0000036045) or empty vector pLKO.1 (SHC001) were purchased from Sigma-Aldrich. Lentivirus productions and lentiviral infections were performed according to the supplier's protocol. After puromycin selection to generate stable cell lines with empty vector shRNA as a control and hST8Sia I-specific shRNA, cells were cultivated for 24 h in FBS-free medium and the knock-down efficacy of hST8Sia I shRNA was assessed by real-time qRT-PCR and Western blot analysis.

# Results

## Serum starvation influences the gene expression of ganglioside-specific human sialyltransferases

It is well known that seven kinds of human sialyltransferases (hB4Gal T I, hB3GalT IV, hST3Gal II, hST3Gal III, hST3Gal V, hST8Sia I, hST8Sia V) are involved in ganglioside biosynthesis [11, 12]. Because we showed increased gene expression of hST3Gal V during MG-63 cell differentiation by serum starvation in a previous study [9], the effects of serum starvation on the gene expression of six human sialyltransferases, excluding hST3Gal V, were investigated by RT-PCR. Serum starvation increased the mRNA levels of the five types of human sialyltransferases, except for hST8Sia V. Among them, the expression level of hST8Sia I was the highest, followed by hST3Gal II, but hB4GalT I, hB3GalT IV, and hST3Gal III showed low levels (S1 Fig). Based on this result, the mechanism of upregulation of *hST8Sia I* expression by serum starvation was further investigated in this study.

## Gene expression of hST8Sia I by serum starvation in human osteosarcoma MG-63 cells

In previous studies, we revealed that the expression of BMP-2 and osteocalcin, well-known markers of osteoblastic differentiation, increased in a time-dependent manner by serum starvation in MG-63 cells [9, 10]. In this study, as shown in Fig 1, we verified that the expression levels of BMP-2 and osteocalcin were augmented by serum starvation in MG-63 cells in a time-dependent manner. In parallel, the time-dependent elevation of *hST8Sia I* expression by

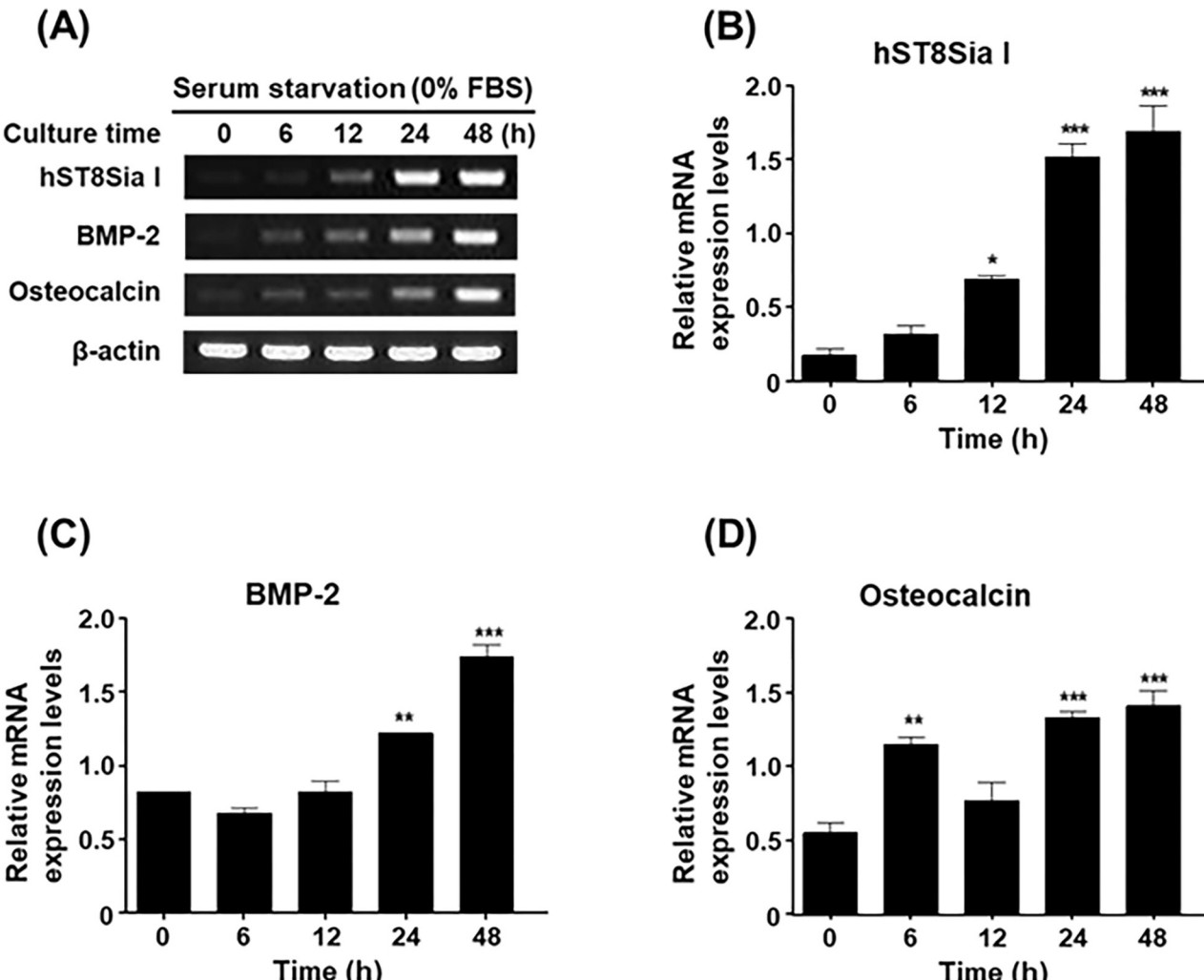

**Fig 1. Effect of SS on expression levels of hST8Sia I and osteoblastic markers.** Total RNA from MG-63 cells was isolated after incubation in serum-free medium for the indicated time periods and mRNA transcripts of hST8Sia I and osteoblastic markers (BMP-2 and osteocalcin) were detected by RT-PCR (A) and quantitative real-time PCR analysis (B-D). As an internal control, parallel reactions were performed to measure levels of the housekeeping gene β-actin. The transcript copy numbers of hST8Sia I and osteoblastic markers were normalized to the β-actin transcript copy number for each sample. Experiments were repeated three times to check reproducibility of results. *$P < 0.05$; **$P < 0.01$. ***$P < 0.001$ (compared with control).

serum starvation was confirmed by RT-PCR and qPCR. These results indicated that the level of *hST8Sia I* expression was upregulated during MG-63 cell differentiation triggered by serum starvation. We observed similar results in the human osteoblastic cell line U2OS (S2 Fig). Furthermore, a distinct elevation of hST8Sia I protein levels in serum starvation-induced MG-63 cells compared with uninduced cells was observed by western blot analysis using an hST8Sia I-recognizing antibody (Fig 2B).

## Ganglioside GD3 expression by serum starvation in human osteosarcoma MG-63 cells

To investigate whether the ganglioside GD3 level produced by hST8Sia I was increased by serum starvation in MG-63 cells, we analyzed ganglioside GD3 expression at the cellular level using an anti-GD3 monoclonal antibody (mAb) and a secondary antibody (FITC-conjugated

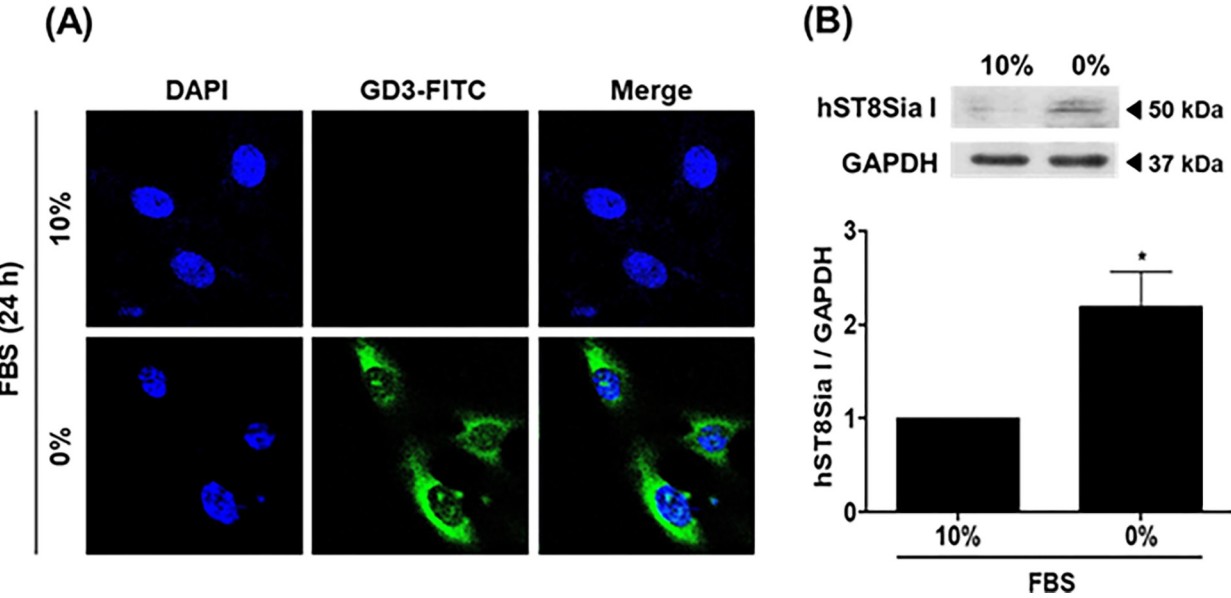

**Fig 2. Confocal analysis of ganglioside GD3 expression and Western blot analysis of hST8Sia I in SS-induced MG-63 cells.** (A) After incubation for 24 h in standard medium containing 10% FBS or 0% FBS, cells were immunostained with anti-GD3 antibodies (FITC; green). Nuclei were stained with DAPI (blue), and analyzed by confocal microscopy (100 ×). (B) Equal amounts of cell lysates (20 μg) were resolved in SDS-polyacrylamide gels and then transferred to PVDF membrane. Membrane was incubated with sequentially with hST8Sia I-specific antibody and horseradish peroxidase (HRP)-conjugated secondary antibody. GAPDH was used as an internal control. The bar graphs represent the intensities of the band obtained by densitometry. The values represent the means ± SEM of three independent experiments with triplicate measurements. *$P < 0.05$.

goat-anti-mouse IgM) by immunofluorescence confocal microscopy. As a result, the increased GD3 expression level was observed in cells cultivated for 24 h in an FBS-free medium, but not in a 10% FBS-containing medium (Fig 2A). This result suggests that the increased gene expression of *hST8Sia I* induced by serum starvation contributes to an elevation of ganglioside GD3 expression at the cellular level.

## Comparison of serum starvation-induced transcriptional activity of hST8Sia I gene promoter in MG-63 cells

To elucidate whether the mRNA levels of *hST8Sia I* were markedly enhanced by its promoter activity induced by serum starvation in MG-63 cells (Fig 1), luciferase reporter plasmids (pGL3-1146/-646 to pGL3-2646/-646) were transfected into MG-63 cells and further incubated for 24 h in medium with or without 10% FBS and analyzed by luciferase assays. As shown in Fig 3A, the four plasmids tested showed significantly increased luciferase activity in cells incubated in an FBS-free medium compared to those incubated in a medium with 10% FBS. Moreover, the luciferase activity obtained with the pGL3-1146/-646 construct revealed the highest increase (approximately 2.3-fold) compared to the other constructs. These results indicate that the nt -1146 to -646 region plays a crucial role in the serum starvation-responsive core promoter of *hST8Sia I* in MG-63 cells.

## Determination of serum starvation-inducible element in the -1146/-646 part of *hST8Sia I* promoter region

Previously, we showed that the nt -1146 to -646 region includes binding sites for transcription factors AP-1, NF-κB, c-Ets-1, and CREB [13–18]. To explore whether these binding sites are

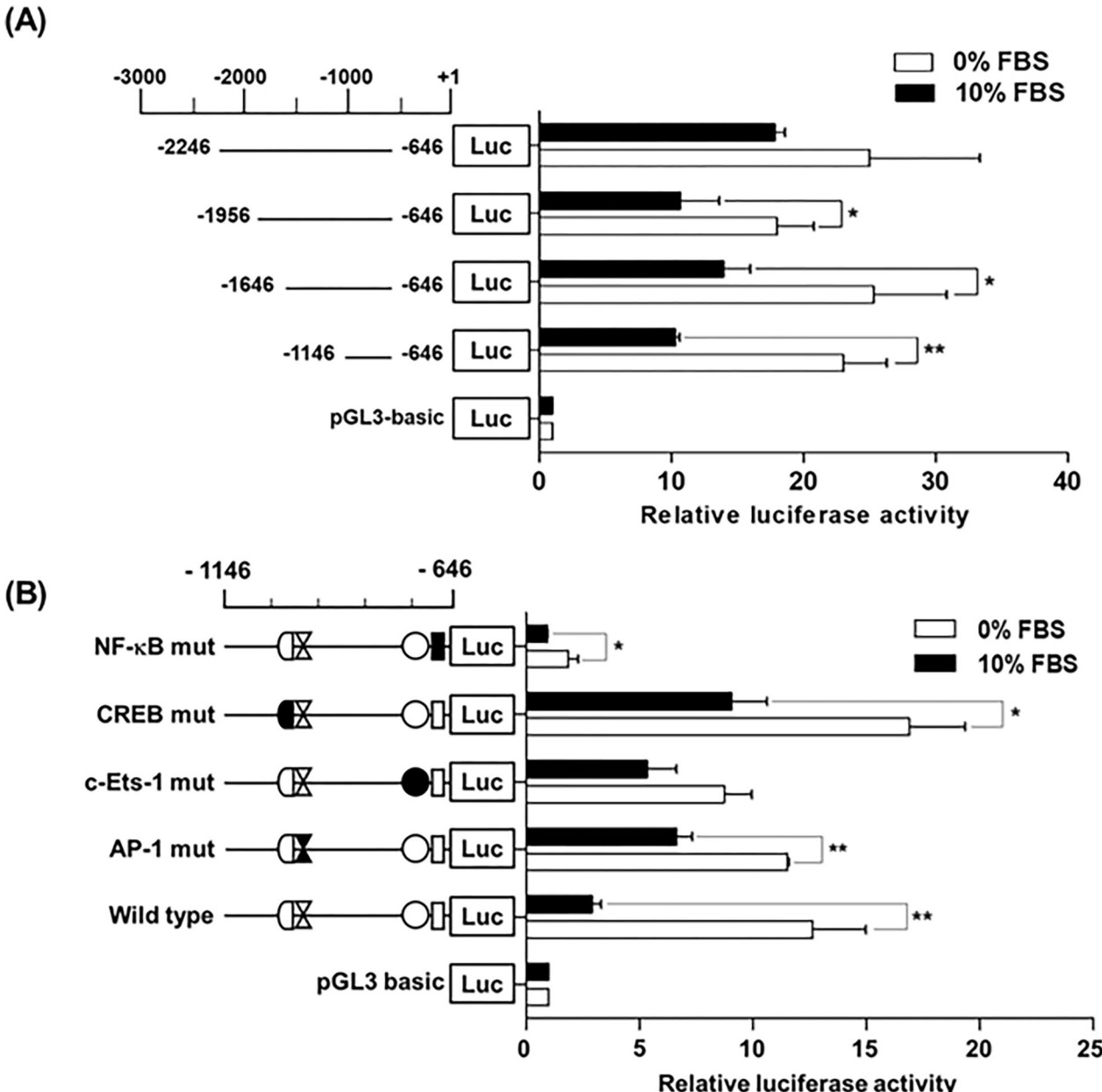

**Fig 3. Analysis of hST8Sia I promoter activity in SS-induced MG-63 cells.** The schematic diagrams represent DNA constructs (A) containing various lengths of the wild-type hST8Sia I promoter, or promoter construct (B) with mutant sequences in the 5'-flanking region, upstream of a luciferase reporter gene; the transcription start site is designated +1. The pGL3-basic construct, which did not contain a promoter or an enhancer, was used as a negative control. Each construct was transfected into MG-63 cells, with pRL-TK co-transfected as an internal control. The transfected cells were incubated in the presence (open bar) or absence (solid bar) of 10% FBS for 24 h. Relative firefly luciferase activity was measured using the Dual-Luciferase Reporter Assay System, and all firefly activity was normalized to the *Renilla* luciferase activity derived from pRL-TK. The values represent the means ± SEM of three independent experiments with triplicate measurements. *$P < 0.05$; **$P < 0.01$.

responsible for serum starvation-inducible expression of *hST8Sia I* in MG-63 cells, four mutant plasmids on these binding sites constructed previously [13–18] were transfected into MG-63 cells and further incubated for 24 h in medium with or without 10% FBS, and analyzed by luciferase assays. As shown in Fig 3B, although mutations in AP-1, CREB, and c-Ets-1 binding sites showed a significant decrease in luciferase activity upon serum starvation compared

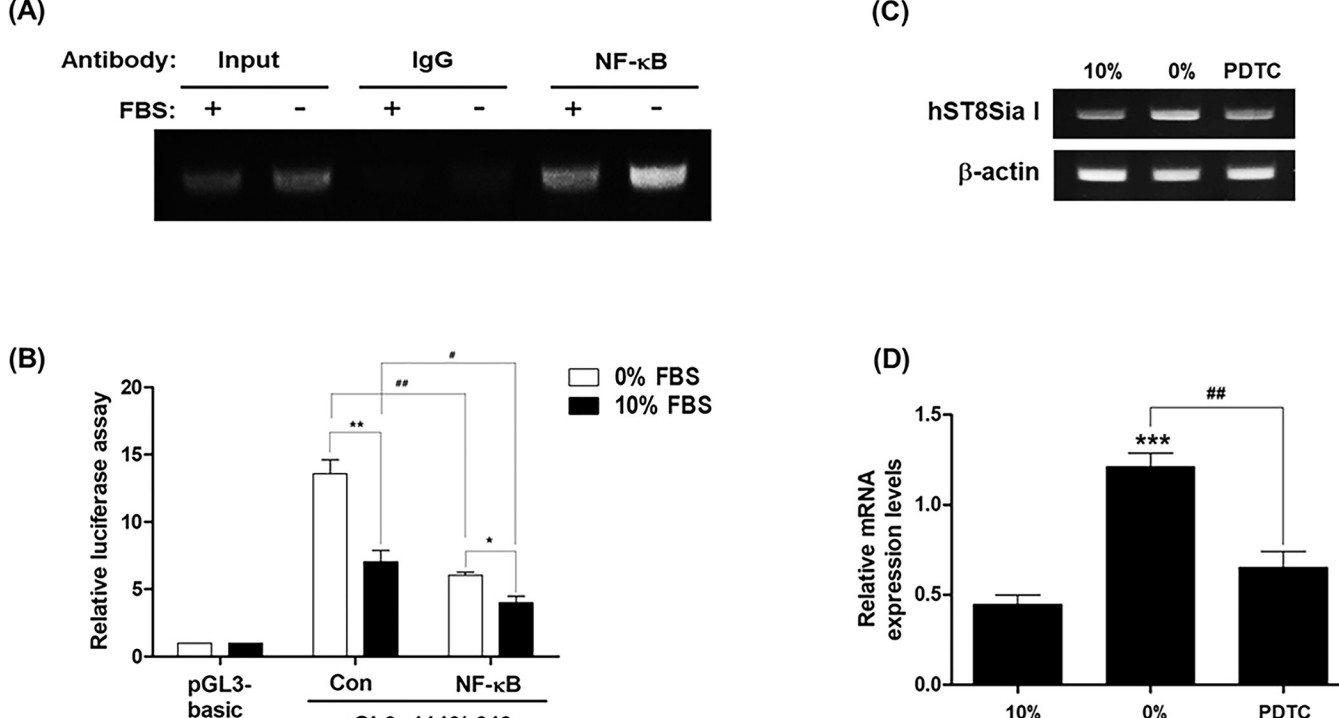

**Fig 4. ChIP analysis and NF-κB inhibition.** (A) ChIP assay was conducted in MG-63 cells grown in the presence or absence of 10% FBS for 24 h with input positive control (10-fold diluted), nonspecific immunoglobulin (IgG), and NF-κB antibody. The -899 and -693 region (207 bp) of the hST8Sia I promoter on immunoprecipitated chromatin obtained from MG-63 cells was amplified by PCR. (B) MG-63 cells were pretreated with 10 μM dithiocarbamate pyrrolidine (PDTC), NF-κB-specific inhibitor, for 1 h and then incubated for 24 h in medium with or without 10% FBS, and analyzed by luciferase assays, RT-PCR (C) and quantitative real-time PCR (D). The values represent the means ± SEM of three independent experiments with triplicate measurements. $^{*}P < 0.05$; $^{**}P < 0.01$. $^{***}P < 0.001$. $^{#}P<0.05$ and $^{##}P<0.01$ compared with 10% FBS and 0% FBS, respectively.

to wild-type pGL3-1146/-646, mutation of the NF-κB binding site almost completely elimi-nated the increased luciferase activity, both in 0% and 10% FBS-containing media, indicating that the binding of NF-κB to its recognition site is essential for transcriptional activation of *hST8Sia I* in MG-63 cells.

To verify that NF-κB was directly related to the serum starvation-induced transcriptional activity of *hST8Sia I* in MG-63 cells, chromatin immunoprecipitation (ChIP) experiments were carried out using MG-63 cells incubated for 24 h in a medium with or without 10% FBS. As shown in Fig 4A, PCR amplification using primers encompassing the NF-κB binding site (-731/-722) showed a marked increase in cells incubated in medium without 10% FBS com-pared to those incubated in medium with 10% FBS. However, a distinct PCR product was not detected in the ChIP assay using the negative control IgG antibody. Moreover, to further con-firm the contribution of NF-κB to serum starvation-induced transcriptional activation of *hST8Sia I* in MG-63 cells, the cells were pretreated with the NF-κB-specific inhibitor dithiocar-bamate pyrrolidine (PDTC) for 1 h and then cultivated for 24 h in medium with or without 10% FBS and analyzed by luciferase assays. As shown in Fig 4B, PDTC treatment markedly decreased the luciferase activity of pGL3-1146/-646 in serum starvation-induced MG-63 cells. Furthermore, repressed expression of *hST8Sia I* by PDTC was observed by RT-PCR (Fig 4C) and qPCR (Fig 4D). Altogether, these results indicate that direct NF-κB binding to its binding site on the promoter region of *hST8Sia I* was enhanced by *hST8Sia I* expression in serum-star-vation-induced MG-63 cells.

## Upregulation of *hST8Sia I* transcription by serum starvation is related to p38MARK pathway in MG-63 Cells

We investigated the signal pathway mediating transcriptional upregulation of hST8Sia I gene by serum starvation in MG-63 cells. As a positive control, promoter activity of pGL3-1146/-646 was elevated in serum starvation-induced MG-63 cells compared to non-induced cells (Fig 5A). This pattern was not significantly changed by GŐ6983 (PKC inhibitor), SP600125 (JNK inhibitor) and U0126 (MEK/ERK inhibitor), whereas LY294002 (PI3K/AKT inhibitor) and compound C (AMPK inhibitor) exerted a slight influence on promoter activity of pGL3-1146/-646 induced by serum starvation. However, SB203580 (p38MAPK inhibitor) treatment resulted in a marked reduction in the luciferase activity of pGL3-1146/-646 in serum starvation-induced MG-63 cells to almost similar level to that in non-induced cells. Furthermore, the repressive gene expression of hST8Sia I by SB203580 was observed by RT-PCR (Fig 5B) and real-time qRT-PCR (Fig 5C). The results have clearly suggested that transcription activity of hST8Sia I gene by serum starvation in MG-63 cells is controlled by p38MAPK pathway. We investigated the signaling pathway mediating the transcriptional upregulation of *hST8Sia I* by serum starvation in MG-63 cells. As a positive control, the promoter activity of pGL3-1146/-646 was elevated in serum starvation-induced MG-63 cells compared to that in non-starvation-induced cells (Fig 5A). This pattern was not significantly changed by GŐ6983 (PKC inhibitor), SP600125 (JNK inhibitor), or U0126 (MEK/ERK inhibitor), whereas LY294002 (a PI3K/AKT inhibitor) and compound C (an AMPK inhibitor) exerted a slight influence on the promoter activity of pGL3-1146/-646 induced by serum starvation. However, SB203580 (p38MAPK inhibitor) treatment resulted in a marked reduction in the luciferase activity of pGL3-1146/-646 in serum-starvation-induced MG-63 cells to a level similar to that in non-

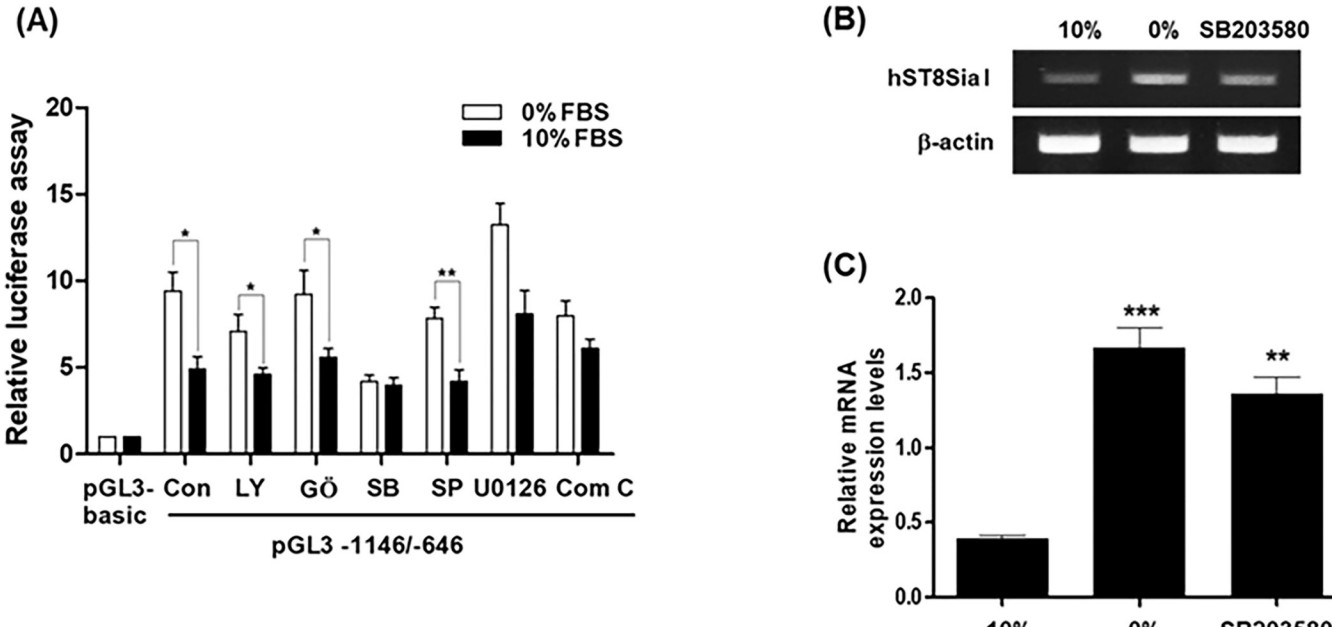

**Fig 5. Effect of SS on signaling pathway controlling transcription of hST8Sia I in MG-63 cells.** (A) The pGL3-basic (negative control), pGL3-1146/-646 (positive control), and pRL-TK (internal control) were co-transfected into MG-63 cells. Transfected cells were incubated in the presence (solid bar) and absence (open bar) of 10% FBS with LY294002 (10 μM), GŐ6983 (100 nM), SB203580 (20 μM), SP600125 (10 μM), U0126 (10 μM), and Compound C (10 μM) inhibitors for 24 h. Relative luciferase activity was normalized with the *Renilla* luciferase activity derived from pRL-TK. MG-63 cells were incubated in the presence and absence of 10% FBS with SB203580 (20 μM) for 24 h and analyzed by RT-PCR (B) and quantitative real-time PCR (C). Data represent mean ± SEM for three independent experiments with triplicate measurements. *P < 0.05; **P < 0.01, ***P < 0.001.

induced cells. Furthermore, repressive gene expression of *hST8Sia I* by SB203580 was observed by RT-PCR (Fig 5B) and real-time qRT-PCR (Fig 5C). These results clearly suggest that the transcriptional activity of *hST8Sia I* by serum starvation in MG-63 cells is controlled by the p38MAPK pathway.

## Effect of ganglioside GD3 on osteoblast differentiation

To investigate whether ganglioside GD3 synthesized by *hST8Sia I* exerts an influence on osteoblast differentiation, *hST8Sia I* was knocked down using shRNA and then the changes in the mRNA levels of *hST8Sia I* and osteoblast differentiation markers, BMP-2, alkaline phosphatase(ALP), and osteocalcin were checked by real-time qRT-PCR analysis. *hST8Sia I* mRNA expression levels were remarkably reduced by 5.7-fold in *hST8Sia I* knockdown cells compared to control cells (Fig 6A), whereas in *hST8Sia I* knockdown cells, the mRNA levels of BMP-2, ALP, and osteocalcin were 1.6-, 2.4- and 4.8-folds higher than those in control cells. In addition, protein levels of *hST8Sia I* and ganglioside GD3 were distinctly decreased in *hST8Sia I* knockdown cells compared to control cells (Fig 6B and 6C). Collectively, these results suggest that expression of *hST8Sia I* and ganglioside GD3 have no apparent effect on the differentiation of MG-63 cells.

## Discussion

Although in human sialyltransferases, gene expression directly associated with ganglioside biosynthesis has been widely reported in embryonic development, inflammation, degeneration, cancer progression, and metastasis [19–22], little is known about their gene expression in

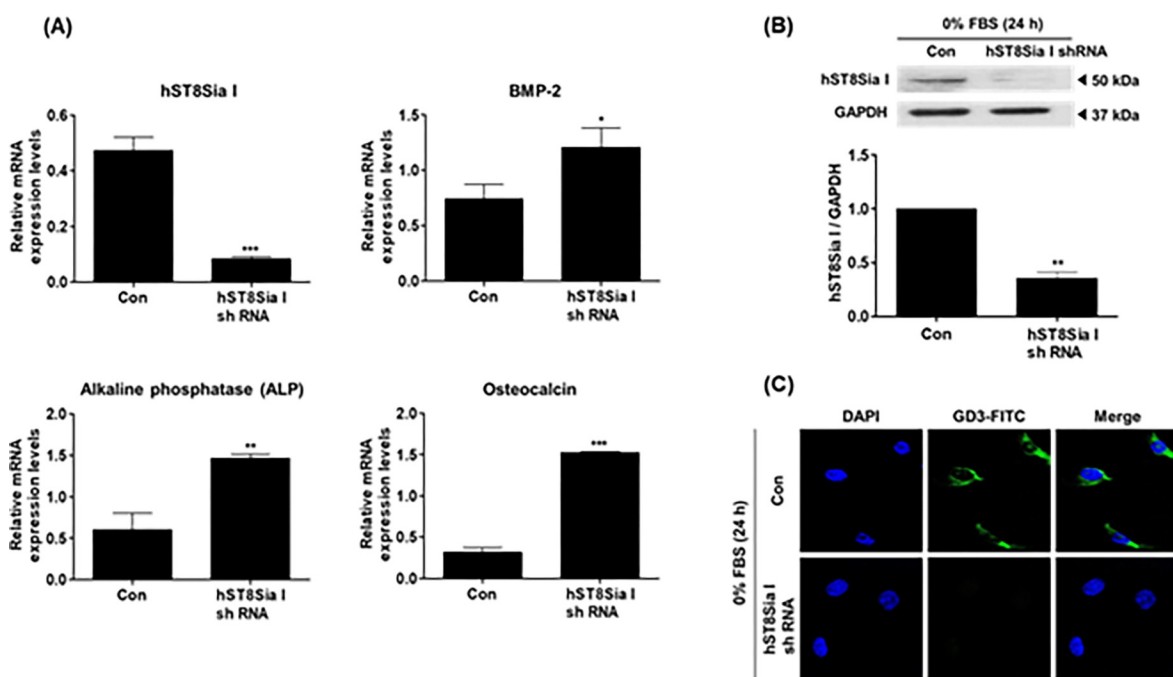

**Fig 6. Knockdown of hST8Sia I using shRNA. MG-63 cells were transfected with empty vector pLKO.1 (control) or hST8Sia I shRNA.** After puromycin, cells were cultivated for 24 h in FBS-free medium. Relative mRNA levels of hSt8Sia I, BMP-2, ALP, and osteoclacin were measured by real time qPCR assays (A). Protein levels of hST8Sia I were analyzed by Western blot assay (B). Data represent mean ± SEM for three independent experiments with triplicate measurements. **$P < 0.01$. After cells were immunostained with anti-GD3 antibodies (FITC; green), ganglioside GD3 expression was analyzed by confocal microscopy (100 ×) (C). Nuclei were stained with DAPI (blue).

relation to osteoblast differentiation. We have previously demonstrated that serum starvation induces cell cycle arrest at the G1 stage preceding cell differentiation of MG-63 cells, but has no significant effect on cell proliferation, with about 73% viability even after serum starvation for 48 h [9]. In this study, we analyzed the expression of six human sialyltransferases (hB4Gal T I, hB3GalT IV, hST3Gal II, hST3Gal III, hST8Sia I, and hST8Sia V) involved in ganglioside biosynthesis during serum starvation. Our RT-PCR results showed that the level of *hST8Sia I* expression was highest in serum starvation-induced MG-83 cells. In our previous studies, we reported for the first time serum starvation-induced osteoblastic differentiation of MG-63 cells and demonstrated that serum starvation significantly elevated the mRNA levels of osteoblast differentiation markers, such as BMP and osteocalcin, along with high expression of *hST6Gal I* and *hST3Gal V* [9, 10]. In line with these observations, we also found that the transcript levels of BMP and osteocalcin increased in a time-dependent manner, along with high mRNA levels of *hST8Sia I*, as demonstrated by RT-PCR and qPCR, suggesting osteoblastic differentiation by serum starvation in MG-63 cells.

In the present study, the effect of serum starvation on osteoblast differentiation *in vitro* was investigated using MG-63 cells. We have demonstrated that expression of *hST8Sia I* at both the mRNA and protein levels was significantly augmented by serum starvation in a time-dependent manner, as confirmed by western blot, RT-PCR, and qPCR analysis, indicating upregulation of *hST8Sia I* by serum starvation at both transcriptional and translational levels in MG-63 cells. Interestingly, gene expression of *hST8Sia I* by serum starvation is clearly similar to that of *hST6Gal I* [10], in that mRNA levels of *hST8Sia I* begin to increase after 6 h and are maximal at 48 h. Moreover, we also demonstrated that elevated *hST8Sia I* expression correlates with increased levels of ganglioside GD3 observed after serum starvation for 24 h in MG-63 cells, as confirmed by immunostaining using GD3 mAb.

In the present study, we clarified that the nt -1146 to -646 promoter region of *hST8Sia I* functions as a serum starvation-responsive core promoter by a luciferase assay using deletion constructs. This region also plays the role of a core promoter crucial for the transcriptional upregulation of *hST8Sia I* in Fas-induced Jurkat T cells [13], human melanoma SK-MEL-2 cells [14], valproic acid-induced SK-N-BE(2)-C human neuroblastoma cells [15], cordycepin-induced SK-N-BE(2)-C human neuroblastoma cells [16], and curcumin-induced A549 human lung cancer cells [17].

Consistently, we also clarified that the NF-κB binding site (-731/-722) in the-1146/-646 region is critical for the transcriptional upregulation of *hST8Sia I* in serum starvation-induced MG-63 cells, as evidenced by site-specific mutagenic analysis, NF-κB-specific inhibitor (PDTC) treatment, and *in vivo* ChIP assay. Of particular interest is our finding that this NF-κB binding site (-731/-722) essentially functions as a major regulator of the transcriptional upregulation of *hST8Sia I* by various extracellular stimuli in different cancer cell types [13–16]. In contrast, we previously confirmed that this NF-κB binding site (-731/-722) mediates transcriptional repression of *hST8Sia I* by triptolide in SK-MEL-2 human melanoma cells [18]. In addition, Bobowski et al. also proved that *hST8Sia I* transcription is suppressed through the inhibition of NF-κB activation by estradiol in human breast cancer cells expressing the estrogen receptor α (ERα) [23].

In previous studies, we have reported that transcriptional upregulation of *hST8Sia I* is caused by Fas signaling in Jurkat T cells [13] and AMPK signaling in A549 human lung cancer cells [17]. In contrast, in the present study, we revealed that the transcriptional upregulation of *hST8Sia I* by serum starvation in MG-63 cells is mediated by the p38 MAPK signaling pathway, as featured by the p38MAPK inhibitor SB203580. Previous studies showed that the p38MAPK signal pathway is involved in osteoblast differentiation [24–27]. Moreover, activation of the p38MAPK pathway triggered NF-κB activation during osteoblastic cell

differentiation [28, 29]. Thus, it may be assumed that the upregulation of *hST8Sia I* transcription by serum starvation in MG-63 cells is mediated by the p38MAPK/NF-κB signaling pathway.

Some studies have reported that ganglioside GD3 plays an important role in cell differentiation [29–32]. For example, it has been reported that GD3 expression by transfection of mouse ST8Sia I cDNA into a murine neuroblastoma cell line, Neuro2a, provoked cell differentiation with neurite sprouting [28, 29]. Our group previously demonstrated that GD3 expression by transfection of *hST8Sia I* cDNA into K562 leukemia cells caused erythroid differentiation [30]. A significant elevation of GD3 expression with GD1a was observed during neural differentiation of human dental pulp stem cells (hDPSCs) [31]. Although it has been reported that during osteoblastic differentiation of human mesenchymal stem cells (hMSCs), suppression of ganglioside GD1a expression by shRNA-mediated knockdown of *hST3Gal II* catalyzing GD1a synthesis caused a remarkable reduction in ALP activity, which suggests the hindrance of osteoblastic differentiation by suppression of GD1a [32], the effect of suppression of GD3 expression on osteoblastic differentiation has yet to be reported. In this study, we investigated for the first time whether suppression of GD3 expression influences osteoblastic differentiation in human osteosarcoma MG-63 cells. Suppression of GD3 expression by knockdown of *hST8Sia I* did not cause any reduction in gene expression levels of BMP, ALP, and osteocalcin known as osteoblast differentiation markers. This result suggests that unlike the effect of GD1a suppression by *hST3Gal II* knockdown, suppression of GD3 expression by *hST8Sia I* knockdown had no apparent influence on osteoblastic differentiation in MG-63 cells.

## Conclusions

Our study proved for the first time that the levels of *hST8Sia I* and ganglioside GD3 are elevated during human osteoblastic MG-63 cell differentiation by serum starvation. Furthermore, we elucidated the transcriptional upregulation of *hST8Sia I* during serum-starvation-triggered MG-63 cell differentiation by promoter analysis using a luciferase reporter assay system. We verified that the -731/-722 sequence, including the NF-κB binding site, is essential for *hST8Sia I* expression induced by serum starvation in MG-63 human osteosarcoma cells by site-directed mutagenesis, NF-κB inhibition, and ChIP assays. Taken together, our results revealed that the transcriptional upregulation of *hST8Sia I* in serum-starvation-induced MG-63 cells is mediated by the p38MAPK/NF-κB signal pathway. The present results also support the conclusion that isolated osteoblasts are sensitive to serum starvation and that serum starvation increases osteoblast activity in that expression of differentiation-related genes in response to serum starvation is increased in cultured MG-63 cells. In addition, our results suggest that the expression of *hST8Sia I* and ganglioside GD3 has no apparent effect on the differentiation of MG-63 cells by shRNA-mediated *hST8Sia I* knockdown.

## Supporting information

**S1 Fig. RT-PCR of six kinds of ganglioside-specific human sialyltransferase genes.** Total RNA from MG-63 cells was isolated after 48 h incubation in medium with or without 10% FBS. (A) mRNA transcripts of six kinds of ganglioside-specific human sialyltransferase were detected by RT-PCR. Housekeeping gene β-actin was used as an internal control. (B) RT-PCR condition and amplified length (bp) were shown.
(TIF)

**S2 Fig. Effect of serum starvation on expression levels of hST8Sia I and osteoblastic markers in human osteosarcoma U2OS cells.** Total RNA from U2OS cells was isolated after 48 h

incubation in serum-free medium for the indicated time period and mRNA transcripts of hST8Sia I and osteoblastic markers (BMP-2 and osteocalcin) were detected by RT-PCR (A). Densitometric intensity of each band was shown (B-D). As an internal control, parallel reactions were performed to measure levels of the housekeeping gene β-actin. Data represent the relative values ± SEM of three independent experiments and the mean values from each experiment were compared using one-way ANOVA. * *P < 0.06* (compared to control).
(TIF)

**S1 File. Original data of western blots (PPT).**
(PPTX)

**S1 Table. Primer sequences used for RT-PCR in this study.**
(TIF)

**S2 Table. Raw data of real time PCR (Excel).**
(XLSX)

## Author Contributions

**Conceptualization:** Jong-Hyun Cho, Hyeon-Jun Kim, Cheorl-Ho Kim, Young-Choon Lee.

**Data curation:** Kyoung-Sook Kim.

**Funding acquisition:** Cheorl-Ho Kim, Young-Choon Lee.

**Investigation:** So-Young An, Hyun-Kyoung Yoon, Hee-Do Kim.

**Methodology:** Kyoung-Sook Kim.

**Project administration:** Cheorl-Ho Kim, Young-Choon Lee.

**Resources:** Jong-Hyun Cho, Hyeon-Jun Kim.

**Supervision:** Cheorl-Ho Kim.

**Validation:** Kyoung-Sook Kim, Hee-Do Kim.

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
