## [Editor Report · Decision Letter 0]

16 Jul 2023

PONE-D-23-19970Upregulation of Human GD3 Synthase (hST8Sia I) Gene Expression During Serum Starvation-Induced Osteoblastic Differentiation of MG-63 CellsPLOS ONE

Dear Dr. Kim,

Thank you for submitting your manuscript to PLOS ONE. After careful consideration, we feel that it has merit but does not fully meet PLOS ONE’s publication criteria as it currently stands. Therefore, we invite you to submit a revised version of the manuscript that addresses the points raised during the review process.

It is not clear how well serum starvation reproduces the impact of nutrient starvation on the organism. The introduction seems too speculative in this regard and does not contribute to the understanding of the experimental design. Serum starvation is a common maneuver to induce a differentiation response in many cell types and there is no need to force a correlation with undernutrition. Moreover, it is difficult to determine whether the impact of serum starvation is due to a nutritional deficiency, oncotic pressure, or growth fact withdrawal without the addition of protein (albumin) in the same quantity provided by serum.

Although NFkB mutation has the largest effect in promoter activity it does not seem to discriminate between 0% and 10% of FBS, suggesting it is not the determining factor in starvation-induced differential expression of hST8Sia. In contrast, the mutation in c-Ets site does not modify basal (10% FBS) expression while abrogating serum starvation-induced expression.  Thus, the sentence "suggesting that NF-κB is implicated in serum starvation-induced transcriptional activity in MG-63 cells" is not supported by the data and should be modified to reflect the role of NF-kB as a major driver of hST8Sia expression and c-Ets as driving the selective higher gene expression in serum-starved cells. The role of NF-kB seems to be the opposite of what was reported for other cell types, however, no possible explanation is provided for it in Discussion.

The crosstalk between NF-kB and p38 signaling pathways in hST8Sia expression could be easily assessed by any of the different assays (gene reporter, ChIP) used in the manuscript. It seems like a missed opportunity, but most importantly figure 7 is speculative without the direct proof that p38 is actually modulating NF-kB activity.

The sentences "Collectively, these results suggest that expression of hST8Sia I and ganglioside GD3 have no apparent effect on the differentiation of MG-63 cells" (page 13) and "Suppression of GD3 expression by knockdown of hST8Sia I did not cause any reduction in gene expression levels of BMP, ALP, and osteocalcin known as osteoblast differentiation markers" does not represent the potential role of hST8Sia I and GD3 in differentiation. The data in Figure 6 point to a negative effect on differentiation markers in particular on osteocalcin expression. Any other effect of serum starvation such as G1 arrest is impacted by the knockdown of hST8Sia? Knockdown of hST8Sia has any impact on the expression of other sialyltransferases?

We look forward to receiving your revised manuscript.

Kind regards,

Bruno Lourenco Diaz, Ph.D.

Academic Editor

PLOS ONE

“Dong-A University research fund”

“This work was supported by the Dong-A University research fund. Article processing charge will be paid by the DAU fund.”

“Dong-A University research fund”
---

## [Author Response · Author response to Decision Letter 0]

30 Aug 2023

Dear Editorial office and dealing editor

I have answered to your questions in item by item in my response letter,and funding source, as I have obtained and can pay for APC.

Thank you

Sincerely

Cheorl-Ho Kim PhD Professor

Editorial Member of PLOS One

---

## [Editor Report · Decision Letter 1]

2 Oct 2023

PONE-D-23-19970R1Upregulation of Human GD3 Synthase (hST8Sia I) Gene Expression During Serum Starvation-Induced Osteoblastic Differentiation of MG-63 CellsPLOS ONE

Dear Dr. Kim,

I am sorry for the delay in your manuscript, but I could not get a response from the editorial office on how to replace the manuscript file with the correct version. I believe the best way to proceed is to send it back to you for revision so you can attach the correct version. Please let me know if you run in any further complications. Thanks for your patience.

We look forward to receiving your revised manuscript.

Kind regards,

Bruno Lourenco Diaz, Ph.D.

Academic Editor

PLOS ONE
---

## [Author Response · Author response to Decision Letter 1]

3 Oct 2023

Dear Editorial office and Editor

I am sorry for my poor uploading our revised manuscript that was uploaded on 30th August, 2023 

As you have kindly suggested, I am reuploading the revised manuscript for your final decision.

For the APC payment, please send me the invoice to 

Dr Yung-Choon Lee, Department of Medicinal Biotechnology, College of Health Sciences, Dong-A university, Busan 604-714, South Korea. Tel: +82-51-200-7591; Fax: +82-51-200-6536. (yclee@dau.ac.kr)

You can also send the invoice to me and I will forward it to him.

Thank you

Sincerely

Cheorl-Ho Kim

---

## [Editor Report · Decision Letter 2]

10 Oct 2023

Upregulation of Human GD3 Synthase (hST8Sia I) Gene Expression During Serum Starvation-Induced Osteoblastic Differentiation of MG-63 Cells

PONE-D-23-19970R2

Dear Dr. Kim,

We’re pleased to inform you that your manuscript has been judged scientifically suitable for publication and will be formally accepted for publication once it meets all outstanding technical requirements.

Kind regards,

Bruno Lourenco Diaz, Ph.D.

Academic Editor

PLOS ONE
---

## [Editor Report · Acceptance letter]

25 Oct 2023

PONE-D-23-19970R2 

Upregulation of Human GD3 Synthase (*hST8Sia I*) Gene Expression During Serum Starvation-Induced Osteoblastic Differentiation of MG-63 Cells 

Dear Dr. Kim:

I'm pleased to inform you that your manuscript has been deemed suitable for publication in PLOS ONE. Congratulations! Your manuscript is now with our production department. 

Kind regards, 

on behalf of

Dr. Bruno Lourenco Diaz 

Academic Editor

PLOS ONE